# Diversity of the Bacterial and Viral Communities in the Tropical Horse Tick, *Dermacentor nitens*, in Colombia

**DOI:** 10.3390/pathogens12070942

**Published:** 2023-07-16

**Authors:** Andres F. Holguin-Rocha, Arley Calle-Tobon, Gissella M. Vásquez, Helvio Astete, Michael L. Fisher, Alberto Tobon-Castano, Gabriel Velez-Tobon, L. Paulina Maldonado-Ruiz, Kristopher Silver, Yoonseong Park, Berlin Londono-Renteria

**Affiliations:** 1Department of Entomology, College of Agriculture, Kansas State University, Manhattan, KS 66506, USA; aholguin@ksu.edu (A.F.H.-R.); lpmaldonado@ksu.edu (L.P.M.-R.); ksilver@ksu.edu (K.S.); 2Grupo Entomologia Medica, Facultad de Medicina, Universidad de Antioquia, Medellin 050010, Colombia; credoest@gmail.com; 3U.S. Naval Medical Research Unit No. 6 (NAMRU-6), Bellavista, Lima 15001, Peru; gissella.m.vasquez.ln@health.mil (G.M.V.); astetehelvio@gmail.com (H.A.); michael.l.fisher2@navy.mil (M.L.F.); 4Grupo Malaria, Facultad de Medicina, Universidad de Antioquia, Medellin 050010, Colombia; alberto.tobon1@udea.edu.co (A.T.-C.); gabrielj.velez@udea.edu.co (G.V.-T.); 5School of Public Health and Tropical Medicine, Tulane University, New Orleans, LA 70112, USA

**Keywords:** next-generation sequencing, metatranscriptomics, 16s rRNA, RNA-seq, *Francisella*-like endosymbiont

## Abstract

Ticks are obligatory hematophagous ectoparasites that transmit pathogens among various vertebrates, including humans. The microbial and viral communities of ticks, including pathogenic microorganisms, are known to be highly diverse. However, the factors driving this diversity are not well understood. The tropical horse tick, *Dermacentor nitens*, is distributed throughout the Americas and it is recognized as a natural vector of *Babesia caballi* and *Theileria equi*, the causal agents of equine piroplasmosis. In this study, we characterized the bacterial and viral communities associated with partially fed *Dermacentor nitens* females collected using a passive survey on horses from field sites representing three distinct geographical areas in the country of Colombia (Bolivar, Antioquia, and Cordoba). RNA-seq and sequencing of the V3 and V4 hypervariable regions of the 16S rRNA gene were performed using the Illumina-Miseq platform (Illumina, San Diego, CA, USA). A total of 356 operational taxonomic units (OTUs) were identified, in which the presumed endosymbiont, Francisellaceae/*Francisella* spp., was predominantly found. Nine contigs corresponding to six different viruses were identified in three viral families: Chuviridae, Rhabdoviridae, and Flaviviridae. Differences in the relative abundance of the microbial composition among the geographical regions were found to be independent of the presence of *Francisella-*like endosymbiont (FLE). The most prevalent bacteria found in each region were *Corynebacterium* in Bolivar, *Staphylococcus* in Antioquia, and *Pseudomonas* in Cordoba. *Rickettsia*-like endosymbionts, mainly recognized as the etiological agent of rickettsioses in Colombia, were detected in the Cordoba samples. Metatranscriptomics revealed 13 contigs containing FLE genes, suggesting a trend of regional differences. These findings suggest regional distinctions among the ticks and their bacterial compositions.

## 1. Introduction

Ticks are important vectors of pathogens that cause livestock and human diseases, such as ehrlichiosis, borreliosis (Lyme disease), human and cattle babesiosis, and theileriosis. Tick-borne encephalitis virus, Powassan virus, and Crimean–Congo hemorrhagic fever virus are some of the most prevalent tick-borne viral infections [1,2]. The risk of emerging and re-emerging tick-borne diseases remains a continuing threat since prevention and management are hampered by suboptimal diagnostics, lack of treatment options for emerging pathogens, and a scarcity of vaccines [3,4]. Further, the increased movement of ticks due to human activities and globalization has been described as a direct factor driving the migration and colonization of human and animal hosts by ticks and their associated pathogens [5]. In addition, global climate change caused by human activities has increased the incidence and diversity of pathogens in various habitats [6].

Ticks harbor diverse microorganisms, including symbionts, in addition to pathogenic organisms, which may have direct positive or negative effects on the tick or other members of their microbial communities [1,7,8]. The interaction between the different bacterial members of the tick’s microbial community is considered an important factor in the transmission of human and animal pathogenic organisms [9,10]. Among non-pathogenic communities, common bacterial endosymbionts found in ticks are mainly associated with *Rickettsia*, *Coxiella*, and *Francisella* genera [1,11,12]. These microorganisms act as primary endosymbionts providing essential nutrients involved in survival, development, and tick fitness, such as the biosynthesis of B vitamins and cofactors such as riboflavin, folic acid, and biotin [13]. Tick endosymbionts are generally tissue specific, with microbial guilds well established in salivary glands, gut, ovaries, and other tissues [14]. Some of these microorganisms, including pathogenic and non-pathogenic bacteria, can be transovarially transmitted to tick offspring [15]. Given the importance of ticks as vectors of many pathogens, understanding ticks and their symbiont compositions in different ecological systems has arisen as an important area of study [2].

The tick microbiome includes communities of viruses, bacteria, protozoa, and fungi [8,14]. Recent experimental approaches to characterize the bacterial diversity in various species of ticks used next-generation sequencing (NGS) of the 16S rRNA gene sequence amplicons [16,17,18]. These studies revealed tick bacterial communities, including mammalian pathogens, that are dependent on the tick species, type of host, and geographic location [4,11,19]. Characterizing tick populations by microbial diversity, may give us a better understanding of the potential intra- and interspecific microbial interactions occurring within the tick host and their involvement in the tick–vector competence of important human and animal diseases [4,7,20].

Viruses are present in all domains of life, and are particularly rich in the phylum Arthropoda, which includes ticks [21]. Metatranscriptomics are a specific set of widely used tools to investigate RNA viruses in ticks. Despite considerable insights into bacterial diversity, our understanding of tick-associated viruses is still limited, and largely unexplored compared with bacterial diversity [22]. Virome studies of ticks collected in Asia, Europe, and North America have revealed the emergence of novel pathogenic tick-borne viruses as well as a dearth of data on tick viromes which point to a strong need for increased viral surveillance and discovery in this group of arthropods [23,24,25]. Progress in sequencing technology and metagenomics data has allowed for an approximation of the viral community composition present in an increasing, but still few, number of tick species [22,24,26,27,28,29,30]. More information from different species may be an efficient strategy to mitigate the increasing threats of tick-borne diseases to human and animal health [2,3,25,30].

The tropical horse tick, *Dermacentor nitens*, is distributed throughout the Americas and it is recognized as a natural vector of *Babesia caballi* and *Theileria equi*, the causal agents of equine piroplasmosis [31,32]. *Dermacentor nitens* is a one-host tick, with three to four generations per year [33]. Severe infestation in vertebrate animals can cause lesions, especially in the ears, and predispose the host to secondary bacterial infections [34]. Although equines are the primary host, natural infestations have been reported in other domestic and companion animals, as well as wild animals [35,36,37]. *Dermacentor nitens* is considered a sporadic ectoparasite of humans, where tick infestations are probably a consequence of humans continually entering or working in infested livestock environments, resulting in a transference of ticks from the host animals to persons [38]. Accidental infestations by *Dermacentor nitens* in humans, related to agricultural activities, may represent a potential danger, although the vectorial capacity of *Dermacentor nitens* for pathogens related to public health is unknown. There are a few studies that document the occurrence of human pathogenic agents in this tick species [39,40].

This study aimed to survey the diversity of microbial communities of *D. nitens* in three distinct geographical regions in Colombia where equine production is common and identify the main bacterial and viral community members present in the ticks using 16S rRNA gene sequences combined with metatranscriptomic. These results will provide a better understanding of an important equine disease vector through the contribution of large numbers of sequences annotated as tick viruses and operons of *Francisella*-like endosymbionts (FLE) and will help to reveal a trend of differences among the three key geographical regions in Colombia.

## 2. Materials and Methods

### 2.1. Sample Collection and Nucleic Acid Extraction

Tick collection was carried out using a passive survey at “La Rinconada” slaughterhouse (06°11’26.0” N; 75°22’43.4” W) in the municipality of Rionegro, Antioquia, Colombia in July and September 2019. A total of 45 blood-fed *D. nitens* adults were obtained from three horses native to each region, Bolivar, Antioquia, and Cordoba (Appendix A). The three departments are located in the northwest of Colombia and share borders with the department of Antioquia. Live ticks were transported to the Universidad de Antioquia facilities, where taxonomical identification was carried out using morphological keys [41], and specimens were subsequently stored at −0 or −80 °C until they could be shipped to Kansas State University. Blood-fed female *D. nitens* collected from horses were pooled and processed based on the host (individual animal) and region (Bolivar, Antioquia, and Cordoba). A total of three horses per region (5 ticks per horse × 3 horses × 3 regions = 45 ticks) were chosen by using a random selection method. The 5 ticks collected from each horse were pooled for DNA and RNA extraction, for a total of 9 pools. Genomic DNA and RNA were extracted independently following manufacturer instructions using Zymo™ DNA and RNA extraction kits (Zymo Research, Irvine, CA, USA) from the pools previously separated from the tick exoskeleton.

### 2.2. NGS Library Preparations and Data Processing

The genomic DNA extracted from each of the 9 pools of ticks was sent to the Genome Sequencing Core at the University of Kansas, Lawrence, KS. Amplicon libraries were prepared by Illumina Miseq targeting the V3-V4 region with the primers 16S-F (5′-TCGTCGGCAGCGTCAGATGTGTATAAGAGACAGCCTACGGGNGGCWGCAG-3′) and 16S-R (5′- GTCTCGTGGGCTCGGAGATGTGTATAAGAGACAGGACTACHVGGGTATCTAATCC-3′) of the 16S rRNA, with an expected length of ~465 basepair (bp) for the DNA analysis [16].

The 16S rRNA sequences were analyzed with Mothur v.1.45, according to the MiSeq Standard Operating Procedure [42]. Operational Taxonomic Units (OTUs) with 97% of identity were clustered and classified using the database SILVA v.138. Raw reads were filtered to a maximum length of 465 base pair without ambiguous bases [43]. Another filtering step was carried out in Excel to remove low-count OTUs with a prevalence in samples of less than 0.005% [44]. Bacterial relative abundance was analyzed in R studio (vegan package), and GraphPad Prism 9.2.0 software [45,46,47]. We also compared the differences in the proportion of the bacterial composition of the regions through a Non-Metric Multidimensional Scaling (NMDS) ordination plot. It is important to note that there is the potential for low-frequency background noise in this dataset due to the absence of blank extraction control during the nucleic acid extraction and bioinformatics workflow [44].

RNA-seq library preparation was carried out with the NEB Next Stranded RNA library kit without PolyA selection of the mRNA. The nine extracted pools of RNA were sent to the Genome Sequencing Core at the University of Kansas, Lawrence, Kansas. For the metatranscriptomics analysis, the RNA-seq reads were processed for removal of Illumina adaptor sequences, trimmed, and quality-based filtered using Fastp software v.0.20.0 [48,49]. The high-quality reads (Phred-score > 30) were removed by mapping onto the reference genome of *Dermacentor silvarum* (assembly ASM1333974v1) and *Equus caballus* (assembly EquCab3.0) using STAR v.2.7 [50]. The unmapped reads (Appendix A) were used to perform the assembly and annotation of the transcriptome by using Trinity and Blast2GO suite in OmicsBox v.2.0.36 software [51,52,53]. Contigs annotated in Blast2GO were reexamined manually by BLASTn and BLASTx (https://blast.ncbi.nlm.nih.gov/Blast.cgi, Accessed on 25 August 2022) to confirm the results and eliminate potential false positives. Empirical Bayes estimation and Fisher’s exact tests (α = 0.05) by pairwise comparison based on the negative binomial distribution analysis were carried out with edgeR by using the Galaxy platform to test statistically significant differences in abundance between the bacterial and viral sequences annotated with the geographic location for the blood-fed *D. nitens*.

### 2.3. Phylogenetic Analyses of Viral, Rickettsia spp., and Francisella spp. Contigs

Phylogenetic analyses by comparison of Bayesian inference, maximum likelihood, minimum evolution, and neighbor-joining methods were performed as an initial assessment for the identification and classification of the bacterial protein sequences and the OTUs detected in this study compared to the reference sequences retrieved from the NCBI GenBank database by conducting homology-based taxonomic assignment and gene function via BLAST. Bacterial protein sequences, partial 16s rRNA nucleotide sequences of *Rickettsia*-like endosymbiont (RLE), FLE, and viral protein sequences were retrieved from the GenBank database as indicated with the GenBank accession numbers in Figures 2–4. Sequences were aligned by using Muscle in MEGA-X software [54]. Bayesian inference analysis was carried out using BEAST v1.10.4 software [55]. Phylogenetic trees for the analysis of the 16s rRNA nucleotide sequences were constructed based on the neighbor-joining method with a pairwise deletion. The trees for the V3–V4 regions sequenced in this study were constructed with 500 bootstrap replicates [56,57,58] unless otherwise specified. For the metatranscriptomic analyses of the FLE and viral protein sequences, the cladograms were constructed using annotated and concatenated genes for each contig by using the maximum likelihood method with the Tamura–Nei model and 500 bootstrap replicates [59].

### 2.4. Ethical Approval

This study was approved by the Bioethics Committee of the Universidad de Antioquia (Approval record No. 15-32-436 of June 2015). It was also granted an environmental license by the Colombian government through the National Environmental Licensing Authority (Autoridad Nacional de Licencias Ambientales-ANLA, Resolution ANLA 00908 of 27 May 2017).

## 3. Results 

### 3.1. Bacterial Diversity Investigated Using V3–V4 Regions of the 16S rRNA Sequences

A total of 372,493 sequences after filtering 392,819 raw reads were assembled into 6686 contigs and assigned to 356 OTUs with a threshold of 97% of sequence identity (Table 1). Notably, the sequences consisted of three main OTUs, all identified as FLE (>80%) in all nine samples (Figure 1A). Among the remaining <20% OTUs, the most prevalent bacteria in different regions were *Corynebacterium* in Bolivar, *Staphylococcus* in Antioquia, and *Pseudomonas* in Cordoba (Figure 1B). We also compared the differences in bacterial compositions of the regions through Non-Metric Multidimensional Scaling (NMDS) in the datasets before and after excluding FLE (Figure 1C,D). Our NMDS plots suggest that regional bacterial composition is unique and independent of the presence of FLE and can be useful to differentiate the bacterial composition from different geographical regions (Figure 1).

The FLEs categorized by a 97% identity threshold were three different OTUs (OTU001, 002, and 010 in Figure 2A and Appendix A). These sequences are significantly different from each other with 20 nucleotide (nt) mismatches between OTU001 and OTU002, 21 nt mismatches between OTU002 and OTU010, and 8 nt mismatches between OTU001 and OTU010. High frequencies of the reads for each FLE OTUs, which are in independent libraries, suggest that the three different FLE OTUs are not sequencing artifacts. The cladogram of the FLE sequences showed these three OTU clustered in a branch with the bootstrapping value of 100 (Figure 2A). A single OTU, OTU184, was categorized into *Rickettsia*-like endosymbiont (RLE) in one pool of the Cordoba region. Phylogenetic analysis supports the position of this sequence in the tree clustered with RLE of *Amblyomma latepunctatum* and a clear separation from the pathogenic *Rickettsia*, although the bootstrapping value was 68 (Figure 2B).

### 3.2. Metatranscriptome Containing Viral and Francisella spp. RNA

A total of 152.2 million raw reads were obtained from the nine pools representing the three different regions. After quality trimming and filtering out against *E. caballus* and *D. silvarum* sequences, 92.18 million reads were used for downstream analysis (Appendix A). De novo assembly was conducted using the TRINITY pipeline built in OmicsBox software. After cleaning and filtering, 16.8 million reads were assembled into 81 contigs. Homology-based taxonomic assignment and gene function for each contig was carried out in Blast2Go and using manual BLAST searches.

Thirteen contigs were categorized as FLE, containing presumed independent operons with an average length of 4794 bp. Table 2 represents the length and coverage information, the sequence name, the gene encoded, and the putative gene size for each contig (Appendix A). The highest coverage of the FLE contigs was Contig_ORF_FLE_of_D. nitens_13, which partially encodes the Mechanosensitive ion channel protein MscS with a length of 596 and 1892.14 TPM (transcripts per million reads) (Appendix A). FLE putative operon sequences were submitted to GenBank with the accession numbers contained in the BioProject PRJNA953638.

Six different putative viruses covered by nine viral contigs with an average length of 1749 bp were identified in BLAST searches for the non-redundant protein database of NCBI and the Viral Genomes database. The sequences were manually inspected and annotated for the coding regions. Table 3 shows the viral contigs with the length and coverage information. The highest coverage for the viral contigs was the D. nitens_Colombia_Flaviviridae_Polyprotein_6 contig with a total of 2346.25 TPM with the coverage predominantly higher in the region of Cordoba (Appendix A). The *D. nitens* virus contig sequences were submitted to GenBank with the accession numbers contained in the BioProject PRJNA953638.

### 3.3. Phylogenetic Analyses of Viral and Francisella spp Contigs

Thirteen FLE groups and nine viral contigs identified by metatranscriptomics were further analyzed for their phylogenetic positions. All 13 FLE contigs clustered with other FLEs identified in tick species when rooted in the pathogenic and opportunistic *Francisella* group. The sequences had a 100% bootstrapping value for the tick endosymbiont clade represented by *Amblyomma maculatum* and *Ornithodoros moubata* [60], Figure 3 showing the phylogeny of the concatenated sequences of 13 contigs. The overall similarity was 90% with the FLE of the Ixodidae family represented by *Aamblyomma maculatum*. The green branched clade, containing *Francisella persica*, *Francisella opportunistica*, and *Francisella hispaniensis*, represents the opportunistic pathogens that have been linked as potential causative agents of illness episodes in humans [12,60,61]. The red-branched clusters, shown as the outgroup, are the pathogenic strains of *Francisella tularensis sl*. To show the relationship of the contigs identified with the FLE clade, the sequence named Contig_ORF_FLE_of_D.nitens_1 was used as a representative sequence for the phylogenetic analysis, mainly because all 13 contigs grouped with the tick-endosymbiont clade. The total coverage found for the 13 contigs classified as FLE was 12,515, with contigs 13 and 1 being the most predominant among all pools of samples (Appendix A).

Phylogenetic analysis of nine viral contigs found three different families for the viral species. The genes were capsid protein, glycoprotein, nucleocapsid, polyprotein, and RNA-dependent RNA polymerase (RdRp) (Table 3). Most of the putative viruses were found by identifying genes encoding RdRp with five annotated sequences and classified into two viral families, Chuviridae and Rhabdoviridae. Two different contigs, D. nitens_Colombia_Chuviridae_Glycoprotein_2 and D. nitens_Colombia_Chuviridae_RdRp_5, were grouped into the same family Chuviridae. Based on the sequence similarities and the tree pattern (Figure 4A.B), these contigs are likely presenting two different viruses, although the name of the closely related virus is the same as Changping Tick Virus 2, a virus that has been reported in China and Turkey infecting *Dermacentor* spp. and *Hyalomma asiaticum* ticks [23,24]. These two viruses were found to be more abundant in the region of Antioquia (Appendix A). The Family Rhabdoviridae is represented by five sequences clustered into two putative viruses (Figure 4C,D). Four of them targeting RdRp were grouped in a clade with Blanchseco virus. The remaining sequence was found encoding a nucleocapsid protein and clustered with the American dog tick Rhabdovirus-2. The contig D. nitens_Colombia_Unclassified_Capsid_Protein_1 showed a close relationship with the capsid protein of Xinjiang tick-associated virus-2, a virus sequence that was presumably reported for the first time in the province of Xinjiang in China. This virus remains as unclassified for the family, and it is grouped with other tick viruses found in *Ixodes scapularis* and *Dermacentor variabilis* (Figure 4E). The family Flaviviridae was found to be represented by one contig named D. nitens_Colombia_Flaviviridae_Polyprotein_6 (Figure 4F). This name was assigned due to the high similarity found with a portion of a Flaviviridae polyprotein from *Haemaphysalis longicornis* and *Rhipicephalus microplus* infesting goats [30].

## 4. Discussion

Hard ticks harbor a considerable diversity of bacteria and viruses, of which there are pathogens of concern for humans and domestic animals [2,4,6,8,62,63,64]. A comprehensive survey of tick microorganisms may allow us to uncover the vectorial capacity of ticks for known pathogens and allow for the early identification of emerging pathogenic microorganisms. In addition, these surveys may provide a better understanding of the interactions among microorganisms under different environmental conditions and across geographic regions. Thus, identifying symbiotic microorganisms and their effects on the vectorial capacity of ticks is critical for predicting future outbreaks caused by febrile diseases of unknown etiology [3].

In this study, metatranscriptomics and bacterial 16S rRNA sequencing enriched the sequence database with newly uncovered *Francisella*-like endosymbionts (FLE) and virus genes in the blood-fed *D. nitens* originating from three different geographical areas in Colombia. Differences in the bacterial microbiome composition of ticks collected from animals coming from Bolivar, Antioquia, and Cordoba populations were found in either inclusion or exclusion of the FLE sequences (Figure 1C,D). The NMDS plot for 16S sequences revealed clusters for the tick geographical origin with a unique bacterial assortment. Geographically separated populations of ticks have previously been shown to have distinctive microbial communities in a number of tick species [17,39,40,65,66]. Microbial community composition could be influenced by other factors, such as the degree of tick engorgement, which has been reported previously [67,68,69]. The capacity of ticks to acquire and spread pathogens may also be significantly impacted by these variations in the microbial community composition.

We found that the most abundant bacterium was FLE (80% of classified reads), which is phylogenetically distantly related to the pathogenic bacteria F. tularensis, and causes tularemia in humans [9]. While Dermacentor variabilis and Dermacentor andersoni are known to carry this pathogen but are distributed in the northern hemisphere where *F. tularensis* is not commonly found, the effect of FLE interactions with pathogens and their role in disease transmission remains unknown [1,11,17,70,71]. Previous results have shown a positive association of vertically transmitted FLE against pathogenic Francisella novicida artificial infection in Dermacentor andersoni, however, Francisella novicida is not considered a tick-borne pathogen, which means this interaction is unlikely to happen under natural conditions [7].

Our results show that the microbial community composition of *D. nitens* appears to vary depending on the geographic location of the species’ population. We observed a higher proportion of FLE in these communities compared to data previously reported for *D. variabilis* (62%), and *D. occidentalis* (41%) in the Americas [17,72]. The highly abundant FLE found in *D. nitens* was a similar finding when compared with previous 16S rRNA sequencing studies on whole-body samples obtained from partially or fully engorged adult *Dermacentor* spp. females as *D. variabilis*, *D. marginatus*, *D. reticulatus*, *D. silvarum*, and *D. albipictus* [72,73,74,75]. Metatranscriptomic analysis suggested high levels of FLE coverage (i.e., transcript per million reads TPM) for Cordoba samples, but without statistical significance in all pairwise comparisons by Student’s t-test. The 16S rRNA analysis, showing the relative abundance, also suggested that the Cordoba population is richer in FLE.

Bolivar, Cordoba, and Antioquia have tropical climates, but with variations. Bolivar is warm and humid, and is known for its coastal areas. Cordoba has distinct wet and dry seasons, with challenges such as heavy rainfall and water availability. Antioquia has diverse climates, with milder conditions in lower areas and cooler temperatures in the mountains. These differences affect agricultural practices, and sustainable land and water management are crucial for addressing environmental challenges [28,29,76,77]. The department of Cordoba, an agricultural stronghold in northern Colombia, has a constant flow and exchange of animals. Thus, the associated ticks may be exposed to a more diverse bacterial environment, which may explain the increased detection frequency of main endosymbiont and transient bacteria, through mechanisms such as horizontal transfer [1,65].

These tendencies of small differences in the communities of endosymbionts related to the geographical origin of the ticks have also been reported for *Dermacentor occidentalis* [17]. In other tick species, such as *Ixodes scapularis*, the endosymbiont population has been shown to impact pathogen infection processes. An unaltered intestinal microbiota favored colonization by *Borrelia burgdorferi s.l.*, while an induced microbial dysbiosis environment showed a negative effect by blocking the colonization of *Anaplasma phagocytophilum* [1,19]. In *Dermacentor nitens*, the transmission of human pathogens is still unknown. However, *D. nitens* ticks collected from equines in Brazil were found positive for *B. burgdorferi s.l.*, the complex known as the causal agent of Lyme disease in the Americas [78]. While *D. nitens’* potential as a Lyme disease vector, and the roles of FLE populations have not been documented, the initial characterization of these FLE populations may provide insights into their involvement in tick–vector competence.

Our FLE sequence analysis revealed three different *D. nitens* FLE variants, OTU001, 002, and 010, with relatively large variations (8 to 21 bp or 1.7 to 4.5% difference) in the V3–V4 region. The sources of these variants are likely from different strains that occur in all three geographical locations. While the genus *Francisella* contains three 16S rRNA copies, we exclude the possibility of intra-genomic variation from these copies based on a study that described a 99.65% minimum similarity average in 1374 Proteobacteria genomic sequences of 16S rRNA [79]. These results are comparable to our previously reported study in *Amblyomma americanun*, where at least two different strains of *Coxiella*-like endosymbionts were found, at the individual tick level [44]. Three *D. nitens* FLE OTUs were monophyletic and also grouped with the FLE of other *Dermacentor* FLEs (Figure 2). We found the FLEs of *R. microplus* and *I. scapularis* were also grouped in this clade [80], indicating, first that endosymbionts are more diverse than previously thought, and second that relatively recent independent invasions or transfers of FLEs frequently occur. This could be due to the fact that the FLE initially evolved from the pathogenic *Francisella* species [1,12,13,60,61,63,72,80,81].

Metatranscriptomics revealed several contigs highly similar to viral families. The Rhabdoviridae family was found as the most abundant and common in the pools of all sequences. This group of Rhabdoviridae viruses (Figure 4D) was also reported for different Ixodidae species such as *Rhipicephalus annulatus*, *R. sanguineus*, *Hyalomma marginatum*, *H. asiaticum*, and *D. variabilis* in the United States [23,24,26]. Blanchseco virus (Rhabdoviridae family) was found in one pool of *Amblyomma ovale* ticks infesting cattle and dogs in Trinidad and Tobago [27]. Similarly, we identified Chuviridae-related sequences in the *D. nitens* RNA pools as the second predominant viral family (Figure 4A). Chuviridae is a newly proposed viral family, that constitutes a large monophyletic group, clustering in an intermediate phylogenetic branch between segmented and unsegmented negative-sense RNA viruses identified in ticks, true flies, mosquitoes, cockroaches, and crabs [23]. The most closely related virus to the *D. nitens* virus, found in this study, was previously identified in China (Figure 4A) with a 90.2% (11,275 out of 12,500 bp) nucleotide sequence identity match. This finding of similar viruses in different continents may originate from the historical commerce of animals between nations such as Colombia and China.

We found geographical differences in the Rhabdoviridae family, according to the contig Rhabdoviridae_RdRp, between the Antioquia and Cordoba regions (*p* = 0.02). Interestingly, we found that the sequence coverage for Rhabdoviridae_Nucleocapsid in the third region, Bolivar, is high when compared with those in the other two regions (*p* = 0.03). This frequency data support the discovery of unique viral community compositions for the three different regions (Appendix A). The coverage of the viral gene composition among the ticks in the three different populations showed statistical differences in transcripts classified into the Rhabdoviridae family (Appendix A). A previous study with *R. microplus*, *D. nitens*, and *R. sanguineus s.l.* in the Magdalena Valley and Magdalena/Urabá ecoregions in Colombia reported the presence of Flaviviridae, Rhabdoviridae, Chuviridae, and unclassified viruses [29]. We conclude that the core RNA virome composition appears to be poor compared with the bacterial endosymbiotic communities.

However, we are aware of the limitations of our study; a small number of horses per region were sampled, so we recommend future studies in the same regions should consider including a larger sample size. Another limitation we had is that identifying viruses by using the few preexisting viral sequences in the GenBank may limit the ability to properly identify novel viruses. This sequence-based survey needs further investigation to understand whether those viruses are transiently acquired with the mammalian blood or established and vertically transmitted.

Finally, this study offers a description of the diversity of bacterial and viral communities of partially fed *D. nitens* female ticks collected in animals originating from three Colombian regions based on our 16S rRNA sequences and transcriptomic analysis. In addition to the differentiated geographical populations in the bacterial and viral composition, we also found multiple co-existing strains of FLE and six different viruses in *D. nitens,* which offers the foundation for future studies. A deeper understanding of the microbial and viral communities hosted by ticks can be utilized to develop future measures to mitigate tick pathogen transmission.

## Figures and Tables

**Figure 1 pathogens-12-00942-f001:**
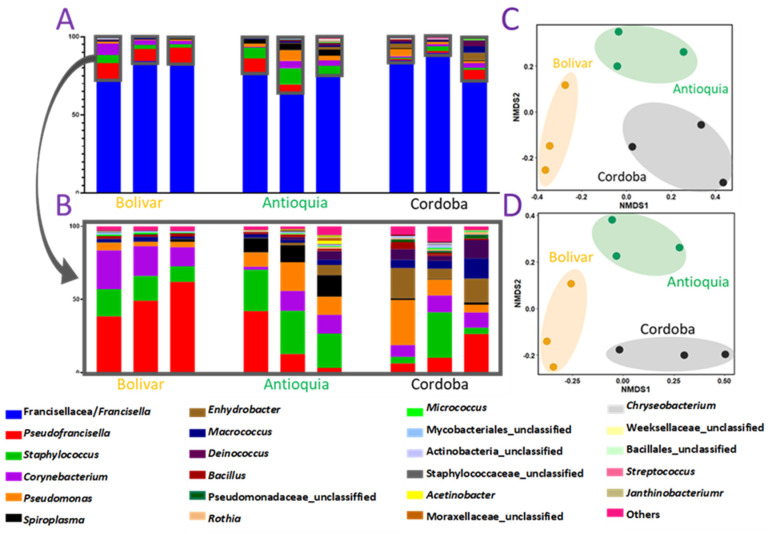
Bacterial diversity shown by genera in 16S rDNA sequences from *Dermacentor nitens* samples collected from three different regions of Colombia. (**A**) Relative abundance is shown by bacterial genera. (**B**) Relative abundance after excluding the sequences of endosymbionts Francisellaceae/*Francisella* spp. (**C**) Non-metric multidimensional scaling plot (NMDS) plot showing the differences among tick samples from different regions. (**D**) NMDS plot showing the differences among tick samples after excluding the endosymbionts.

**Figure 2 pathogens-12-00942-f002:**
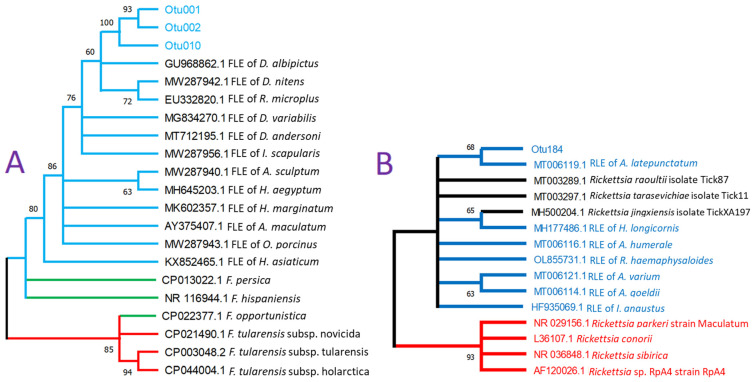
Phylogenetic analyses for the *Francisella-*like endosymbionts (FLE, **A**) and *Rickettsia*-like endosymbionts (RLE, **B**) identified in this study for *Dermacentor nitens* samples. (**A**) Neighbor-joining cladogram rooted to *Francisella tularensis* strains representing the phylogenetic relationship of 16S rDNA sequence (465 bp) OTUs classified as *Francisella* spp. in *Dermacentor nitens*. The tree was built using the pairwise deletion method. Blue branches represent the FLE clade, green branches represent opportunistic pathogenic *Francisella* species, and red branches represent the pathogenic *Francisella tularensis* strains as an outgroup. (**B**) Neighbor-joining cladogram rooted to pathogenic *Rickettsia* strains to represent the phylogenetic relationship of rickettsial 16S rDNA sequences (465 bp) with the OTU184 classified as *Rickettsia* spp. in the *D. nitens* sample. Red branches represent pathogenic *Rickettsia* spp., blue branches represent the sequences of RLE, and dark branches represent candidate–human pathogenic *Rickettsia*. The OTUs were determined by a 97% identity threshold. Bootstrapping percentages in 500 replications are shown on the nodes with a 60% cut-off. The GenBank accession numbers for each sequence are shown at the beginning of names of taxa.

**Figure 3 pathogens-12-00942-f003:**
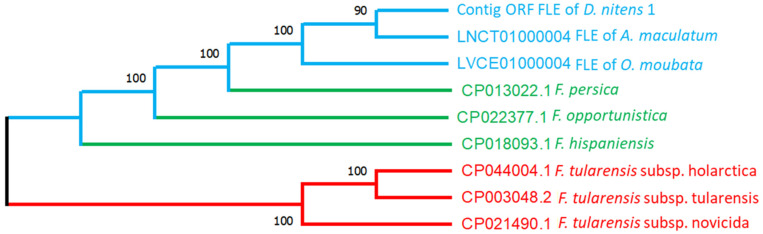
Phylogenetic relationship of the *Francisella*-like endosymbiont in the *D. nitens* samples in this study. The sequence is the translated sequence for the concatenated open reading frames. The selected contig contains nine genes (Table 2) annotated with a total length for the concatenated contig of 3323 amino acids (9969 bp). and 1892 transcript per million (TPM) in the pooled metatranscriptome. The tree is for maximum likelihood cladogram built using the complete deletion method. Bootstrapping percentage values are based on 500 replications and are shown at the nodes. The outgroup is for the sequences of pathogenic *F. tularensis* strains. The blue lines correspond to tick FLE, the green lines correspond to opportunistic pathogens, and the red lines correspond to pathogenic strains of *F. tularensis*. The GenBank accession numbers are shown at the beginning of each label.

**Figure 4 pathogens-12-00942-f004:**
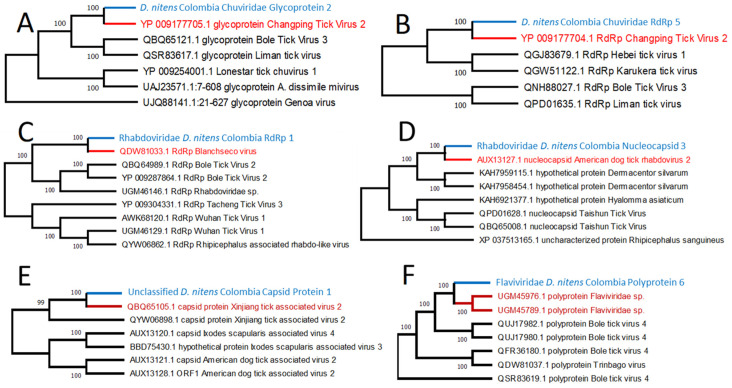
Phylogenetic relationship of the contigs for RNA viruses captured in the *D. nitens* samples in this study. The maximum likelihood cladograms were constructed with complete deletion of assembly gaps. Bootstrapping percentages in 500 replications are shown at the nodes. The contig *D. nitens Colombia Chuviridae Glycoprotein 2* encodes a glycoprotein gene with a length of 668 bp (**A**), *D.nitens_Colombia_Chuviridae_Polymerase_5* encodes an RNA-dependent RNA polymerase with a length of 2156 (**B**), *Rhabdoviridae_Dermacentor_nitens_Colombia_Polymerase_1* encodes an RNA-dependent RNA polymerase with a length of 7061 bp (**C**), *Rhabdoviridae_Dermacentor_nitens_Colombia_Nucleocapsid_3* encodes a nucleocapsid with a length of 524 bp (**D**), *Unclassified_Dermacentor_nitens_Capsid_Protein_1* encodes a capsid protein with a length of 168 bp (**E**), *Flaviviridae_Dermacentor_nitens_Colombia_Polyprotein_6* encodes a polyprotein with a length of 5140 bp (**F**). Names in blue correspond to the viral contigs found in this study, and red names correspond to the closest viral protein sequence in the GenBank database. The GenBank accession numbers are shown at the beginning of the names of taxa.

**Table 1 pathogens-12-00942-t001:** Nine sequencing libraries for the pools for *D. nitens*, targeting V3–V4 regions of the 16 rRNA gene.

Library (Paired Reads)	Region	Raw reads	Mapped Reads	Contigs
DNA_Pool_1	Bolivar	48852	46109	706
DNA_Pool_2	Bolivar	41430	39512	508
DNA_Pool_3	Bolivar	37846	36438	503
DNA_Pool_4	Antioquia	45141	42948	842
DNA_Pool_5	Antioquia	39380	37847	1044
DNA_Pool_6	Antioquia	43778	41116	886
DNA_Pool_7	Cordoba	47878	45604	665
DNA_Pool_8	Cordoba	41244	38268	583
DNA_Pool_9	Cordoba	47270	44651	949
Total		392819	372493	6686

**Table 2 pathogens-12-00942-t002:** Annotations of bacterial contigs captured in the metatranscriptome of *Dermacentor nitens*.

Sequence ID	Gene Name	Open Reading Frame (bp)
Contig_FLE_D.nitens_1, length = 9969 bp, Coverage = 1628		
TRINITY_DN179725_c0_g1_Gene1	3-Oxoacyl-ACP synthase CDS	972
TRINITY_DN179725_c0_g1_Gene2	Phosphate acyltransferase CDS	1047
TRINITY_DN179725_c0_g1_Gene3	rpmF CDS	183
TRINITY_DN179725_c0_g1_Gene4	Hypothetical protein CDS	504
TRINITY_DN179725_c0_g1_Gene5	Transketolase CDS	1992
TRINITY_DN179725_c0_g1_Gene6	Glyceraldehyde-3-phospate dehydrogenase CDS	1002
TRINITY_DN179725_c0_g1_Gene7	Phosphoglycerate kinase CDS	1179
TRINITY_DN179725_c0_g1_Gene8	Pyruvate kinase CDS	1437
TRINITY_DN179725_c0_g1_Gene9	Fructose-1,6-bisphosphate aldolase CDS	1065
Contig_FLE_D.nitens_2, length = 5250 bp, Coverage = 696		
TRINITY_DN15830_c0_g2_Gene1	Nucleotide exchange factor GrpE CDS	588
TRINITY_DN15830_c0_g2_Gene2	Molecular chaperone DnaK CDS	1929
TRINITY_DN15830_c0_g2_Gene3	Molecular chaperone DnaJ CDS	1122
TRINITY_DN15830_c0_g2_Gene4	LysR family transcriptional regulator CDS	906
TRINITY_DN15830_c0_g2_Gene5	Hypothetical protein CDS	705
Contig_FLE_D.nitens_3, length = 8089 bp, Coverage = 675		
TRINITY_DN25174_c0_g1_Gene1	Hypothetical protein CDS	1444
TRINITY_DN25174_c0_g1_Gene2	Hypothetical protein CDS	620
TRINITY_DN25174_c0_g1_Gene3	Hypothetical protein CDS	1006
TRINITY_DN25174_c0_g1_Gene4	Hypothetical protein CDS	1003
TRINITY_DN25174_c0_g1_Gene5	Membrane protein CDS	478
TRINITY_DN25174_c0_g1_Gene6	Hypothetical protein CDS	934
TRINITY_DN25174_c0_g1_Gene7	moxR CDS	962
TRINITY_DN25174_c0_g1_Gene8	Hypothetical protein CDS	444
TRINITY_DN25174_c0_g1_Gene9	pdcY CDS	853
TRINITY_DN25174_c0_g1_Gene10	Hypothetical protein CDS	345
Contig_FLE_D.nitens_4, length = 5373 bp, Coverage = 660		
TRINITY_DN3539_c0_g1_Gene1	Carbamoyl phosphate synthase small subunit CDS	1167
TRINITY_DN3539_c0_g1_Gene2	Carbamoyl phosphate synthase large subunit CDS	3285
TRINITY_DN3539_c0_g1_Gene3	Aspartate carbamoyltransferase CDS	921
Contig_FLE_D.nitens_5, length = 5215 bp, Coverage = 617		
TRINITY_DN112697_c0_g1_Gene1	Coproporphyrinogen III oxidase CDS	1143
TRINITY_DN112697_c0_g1_Gene2	Polysacccharide biosynthesis protein GtrA CDS	378
TRINITY_DN112697_c0_g1_Gene3	Peroxidase CDS	882
TRINITY_DN112697_c0_g1_Gene4	Aconitate hydratase CDS	2812
Contig_FLE_D.nitens_6, length = 1350 bp, Coverage = 787		
TRINITY_DN1678_c0_g1_Gene1	Glutamate dehydrogenase CDS	1350
Contig_FLE_D.nitens_7, length = 2846 bp, Coverage = 942		
TRINITY_DN396500_c0_g1_Gene1	Glycine dehydrogenase CDS	1381
TRINITY_DN396500_c0_g1_Gene2	Glycine dehydrogenase CDS	1465
Contig_FLE_D.nitens_8, length = 4254 bp, Coverage = 880		
TRINITY_DN1569_c0_g1_Gene1	ATP synthase subunit alpha CDS	1542
TRINITY_DN1569_c0_g1_Gene2	ATP F0F1 synthase subunit gamma CDS	897
TRINITY_DN1569_c0_g1_Gene3	ATP synthase subunit beta CDS	1377
TRINITY_DN1569_c0_g1_Gene4	atpC CDS	438
Contig_FLE_D.nitens_9, length = 7945 bp, Coverage = 1393		
TRINITY_DN253568_c0_g1_Gene1	Leucyl aminopeptidase CDS	1440
TRINITY_DN253568_c0_g1_Gene2	lptF CDS	1087
TRINITY_DN253568_c0_g1_Gene3	lptG CDS	1063
TRINITY_DN253568_c0_g1_Gene4	Insulinase family protein CDS	1254
TRINITY_DN253568_c0_g1_Gene5	Insulinase family protein CDS	1254
TRINITY_DN253568_c0_g1_Gene6	rsmD CDS	579
TRINITY_DN253568_c0_g1_Gene7	Trimeric intracellular cation channel family protein CDS	654
TRINITY_DN253568_c0_g1_Gene8	tRNA-(ms [2]io [6]A)-hydrolase CDS	614
Contig_FLE_D.nitens_10, length = 3170 bp, Coverage = 221		
TRINITY_DN182378_c0_g1_Gene1	Amino acid transporter CDS	705
TRINITY_DN182378_c0_g1_Gene2	Oxidoreductase, short chain dehydrogenase/reductase family CDS	827
TRINITY_DN182378_c0_g1_Gene3	Hypothetical protein CDS	471
TRINITY_DN182378_c0_g1_Gene4	NAD(FAD)-utilizing dehydrogenase CDS	1167
Contig_FLE_D.nitens_11, length = 4745 bp, Coverage = 306		
TRINITY_DN15837_c0_g1_Gene1	Hypothetical protein CDS	653
TRINITY_DN15837_c0_g1_Gene2	Hypothetical protein CDS	417
TRINITY_DN15837_c0_g1_Gene3	Alanine--tRNA ligase CDS	2598
TRINITY_DN15837_c0_g1_Gene4	Transporter CDS	1077
Contig_FLE_D.nitens_12, length = 3517 bp, Coverage = 491		
TRINITY_DN182530_c0_g1_Gene1	Hypothetical protein CDS	537
TRINITY_DN182530_c0_g1_Gene2	rpIT CDS	357
TRINITY_DN182530_c0_g1_Gene3	50S ribosomal protein L35 CDS	199
TRINITY_DN182530_c0_g1_Gene4	Translation initiation factor IF-3 CDS	519
TRINITY_DN182530_c0_g1_Gene5	Threonine--tRNA ligase CDS	1905
Contig_FLE_D.nitens_13 length = 596 bp, Coverage = 3219		
TRINITY_DN15777_c0_g1_Gene1	Mechanosensitive ion channel protein MscS-Partial	596
Total coverage	12,515	

**Table 3 pathogens-12-00942-t003:** Viral contigs captured in the metatranscriptome of *D. nitens*, shown for the lengths, coverages, and Blast results.

Contig ID	Length	Coverage	Sequence Name	Blast Result		
				GenBank ID	e-Value	Name of Virus
Unclassified_Capsid_Protein_1	198	1	TRINITY_DN36539_c0_g1	QBQ65105.1	4.00 × 10^−140^	Xinjiang Tick associated virus 2
Chuviridae_Glycoprotein_2	668	168	TRINITY_DN179920_c0_g1	YP_00917 7705.1	0	Changping Tick Virus 2
Chuviridae_Polymerase_5	2156	355	TRINITY_DN180002_c0_g1	YP_009177704.1	0	Changping Tick Virus 2
Rhabdoviridae_Nucleocapsid_3	524	4	TRINITY_DN327528_c0_g1	AUX13127.1	0	American dog tick rhabdovirus 2
Rhabdoviridae_Polymerase_1	7061	218	TRINITY_DN16706_c0_g1	QDW81034.1	0	Blanchseco virus
			TRINITY_DN399801_c0_g1	QDW81033.1	0	Blanchseco virus
			TRINITY_DN405583_c0_g1	QDW81033.1	0	Blanchseco virus
			TRINITY_DN31349_c0_g1	QDW81033.1	0	Blanchseco virus
Flaviviridae_Polyprotein_6	5140	3374	TRINITY_DN544_c0_g1	UGM45976.1	0	Flaviviridae sp.
Total coverage		4120				

## Data Availability

Not applicable.

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
