# Peer review of "Diversity of the Bacterial and Viral Communities in the Tropical Horse Tick, Dermacentor nitens, in Colombia"

_pathogens, 2023, doi:10.3390/pathogens12070942_

Round 1

Reviewer 1 Report

This is an interesting study that is well-written and presents important information.

My main concern is about the methods sections in terms of sampling 3 horses. This is a small number.

Discussion:

Add a small paragraph about the limitations of this study. For example, you sampled 3 horses per region (see line #107) and this is a very small sample size. Thus, higher microbial and viral diversity is expected when sampling a larger number of horses. Therefore, future studies in the same region should consider a larger sample size.

Line 336 >>> sequences. (Figures 1C and 1D).>>> Remove the period after the word sequences.

Are there differences in the weather/environmental conditions of the three regions (Bolivar, Antioquia, and Cordoba) that can be linked to the results of this paper? If yes, try to discuss them or some of them.

Author Response

Response to Reviewer 1 Comments

Point 1: My main concern is about the methods sections in terms of sampling 3 horses. This is a small number.

Response 1: We appreciate your comment. We admit that the sampling, 5 ticks per horse x 3 horses x 3 regions = 45 ticks (Line 119), would be a minimally required number for a comparative study. Although 15 ticks per region representing 3 horses are not sufficient to characterize the populations, the samples revealed valuable information in the metagenomics and microbiome studies. Please consider this report providing the baseline knowledge for an expanded study in the future.

Point 2: Add a small paragraph about the limitations of this study. For example, you sampled 3 horses per region (see line #107) and this is a very small sample size. Thus, higher microbial and viral diversity is expected when sampling a larger number of horses. Therefore, future studies in the same region should consider a larger sample size.

Response 2: We appreciate your understanding and suggestions, a paragraph is now added in the line #438 and after.

“Although the current study provided a baseline for understanding the viral and microbial diversities in three different populations, the interpretation of the study has limitations. The tick sample size is limited in this study for the characterization of local populations. We also may have failed to identify novel viruses because the search relied on preexisting viral sequences in the GenBank data. In addition, further studies are required to conclude whether those viruses are transiently acquired with the mammalian blood or established and transmitted.”

Point 3: Line 336 >>> sequences. (Figures 1C and 1D).>>> Remove the period after the word sequences.

Response 3: Thank you for your suggestion, the period has been deleted and the paragraph now is in the Line #344.

Point 4: Are there differences in the weather/environmental conditions of the three regions (Bolivar, Antioquia, and Cordoba) that can be linked to the results of this paper? If yes, try to discuss them or some of them.

Response 4: We appreciate your comment, definitvely there are differences in the weather/environmental conditions of the three regions. This has been included in the Line #373 and after.

Reviewer 2 Report

Holguin-Rocha et al. have produced a well-constructed study exploring the microbiome and virome of two important pest ticks in Colombia. Although not pathogens per se were detected in this study they do identify and novel Francisella endosymbionts and several novel putative viruses. Whether these viruses are able to infect vertebrate hosts or not (i.e. are tick-specific viruses) needs to be clarified in the discussion. Overall, the introduction is well written and provides the appropriate background. The methods section requires some additional information regarding their phylogenetic methods and perhaps justification of why they used NJ phylogenetic reconstruction and presented cladograms rather than phylograms for much of their study. In addition, many Francisella endosymbionts have been described by their 16S rRNA sequences, which could have easily been sequences here by standard PCR and Sanger sequencing. This would add significant more weight to the paper. Their results are well presented and discussed, although there are numerous grammatical issued which need addressing.

Overall, the authors have presented a nice study which adds important data and resources to the literature and addressing important gaps in our knowledge of tick microbial diversity in south America. Pending revision of the manuscript to address reviewer comments I would be very happy to recommend this work for publication.

Grammatical issues have been highlighted in yellow in the attached annotated pdf and require correction. I recommend editing by an English language expert or native speaker.

# Abstract

Line 28 (and throughout): In my opinion the term "Francisella-like endosymbiont" is an out-dated term used during the initial discovery phase of these bacterial endosymbionts. There is now a wealth of data supporting the inclusion of these endosymbionts into the genus *Francisella*, and therefore there is no need to refer to them as *Francisella*-like. They are simply *Francisella* sp. endosymbionts. I recommend changing this terminology throughout the paper. This is also the case for "*Rickettsia*-like endosymbionts" - they are just Rickettsia endosymbionts.

# Introduction

Line 39: Lyme disease is a form of borreliosis, please correct.

Line 40-41: "one” - requires grammatical correction.

Line 44: "Habitat changes of the ticks" - requires grammatical correction.

Line 49 "in addition to" - requires grammatical correction.

Line 77: The second half of this sentence does not follow from the first half.

Line 81: Please revise sentence to improve meaning and clarity.

# Methods

Lines 133: ggpubr is a figure formatting package and does not need to be cited here.

Line 142: Please provide the proper journal article reference for the BLAST algorithm.

Line 155: Methods for the phylogenetic analysis of Rickettsia sequences are not specified.

Line 157: "initial assessment" of what?

Line 159: "pulled out from” - requires grammatical correction.

Line 163: Was model selection performed prior to phylogenetic analysis?

# Results

Line 208: If this Rickettsia is a true endosymbiont of D. nitens, wouldn't you expect it to be highly abundant in *all* samples, or at least all pools from the one region? In addition, it does not appear that your Rickettsial endosymbiont groups closely with other Dermacentor Rickettsial endosymbionts (Figure 2B). Is it possible that another tick species harbouring this rickettsia was accidently included into this pool?

Figure 2A & B. Please specify how many bp of the 16S rRNA gene was used to construct this tree. Many additional Francisella and Rickettsia sequences occur in GenBank - why were more not added to give a better picture of where these novel spp. fit into the phylogeny. Can you justify why the NJ cladogram is shown rather than the Bayesian or ML phylogram? How did the tree phylogenetic analyses compare? PCR and Sanger sequencing could have been done to generate more informative Francisella and Rickettsia sequences for phylogenetic analysis. Can you justify why this was not done?

Figure 4. Again, why are cladograms presented rather than phylograms?

# Discussion

Line 344: This statement is misleading. Francisella endosymbionts are clearly phylogenetically distinct from F. tularensis.

line 345: Clarify that the ticks, and not F. tularensis is common in the northern hemisphere.

Line 407: Is there evidence that Chuviridae infect vertebrate hosts? If not, this hypothesis may not be correct.

I recommend editing by an English language expert or native speaker. Corrections required throughout. See attached MS.

Author Response

Response to Reviewer 2 Comments

Point 1: Line 28 (and throughout): In my opinion the term "Francisella-like endosymbiont" is an out-dated term used during the initial discovery phase of these bacterial endosymbionts. There is now a wealth of data supporting the inclusion of these endosymbionts into the genus *Francisella*, and therefore there is no need to refer to them as *Francisella*-like. They are simply *Francisella* sp. endosymbionts. I recommend changing this terminology throughout the paper. This is also the case for "*Rickettsia*-like endosymbionts" - they are just Rickettsia endosymbionts.

Response 1: We appreciate your comment, the reason why we have named the groups of endosymbiotic bacteria as "-like endosymbionts" is because this type of nomenclature is still the commonly used method to describe this type of microorganisms. Mainly because they have simply been related to the genus but there is still no valid and accepted description for the species that allows the use of binomial nomenclature with this type of bacteria.

Point 2: # Introduction

Line 39: Lyme disease is a form of borreliosis, please correct.

Line 40-41: "one” - requires grammatical correction.

Line 44: "Habitat changes of the ticks" - requires grammatical correction.

Line 49 "in addition to" - requires grammatical correction.

Line 77: The second half of this sentence does not follow from the first half.

Line 81: Please revise sentence to improve meaning and clarity.

Response 2: Thank you for the editings, all the suggestions were made in the following order and showed with the updated line number:

Line 39: The sentence was corrected and can be found in the updated version as: “Ticks are important vectors of pathogens that cause livestock and human diseases, such as ehrlichiosis, borreliosis (Lyme disease), human and cattle babesiosis, and…”

Lines 40-41: The word “one” was corrected and replaced: with “some”: “Tick-borne encephalitis virus, Powassan virus, and Crimean-Congo hemorrhagic fever virus are some of the most prevalent tick-borne viral infections…”

Lines 44-47: The sentence containing "Habitat changes of the ticks" was modified after English editing by a native speaker and can be found in the updated document as: “Further, increased movement of ticks due to human activities and globalization have been described as direct factors driving migration and colonization of human and animal hosts by ticks and their associated pathogens…”

Line 49:  The sentence containing "in addition to" was edited and can be found now in line 47 in the updated document as: “In addition, global climate change caused by human activities…”

Lines 77-80: Sentence was edited and connected with the first half: “Virome studies of ticks collected in Asia, Europe, and North America have revealed the emergence of novel pathogenic tick-borne viruses as well as a dearth of data on tick viromes which point to a strong need for increased viral surveillance and discovery in this group of arthropods…”

Line 81: The sentence was reviewed and edited to clarify the idea and now can be found in line 83: “More information from different species may be an efficient strategy to mitigate the increasing threats of tick-borne diseases to human and animal health…”

Point 3: # Methods

Lines 133: ggpubr is a figure formatting package and does not need to be cited here.

Line 142: Please provide the proper journal article reference for the BLAST algorithm.

Line 155: Methods for the phylogenetic analysis of Rickettsia sequences are not specified.

Line 157: "initial assessment" of what?

Line 159: "pulled out from” - requires grammatical correction.

Line 163: Was model selection performed prior to phylogenetic analysis?

Response 3: We appreciate your editings, all the suggestions were made in the following order and showed with the updated line number:

Line 133: Edition can be found now in Line 138, ggpubr citation has been removed:

Line 142: The proper reference has been added as reference #49 and can be found in line 148. “Bacterial relative abundance was analyzed in R studio (vegan package), and GraphPad Prism 9.2.0 software”

Line 155: Methods for the phylogenetic analysis of FLE are included in line 162 and after: “…the identification and classification of the bacterial protein sequences and the OTUs detected in this study compared to the reference sequences retrieved from the NCBI GenBank database by doing homology-based via BLAST. Bacterial protein sequences, partial 16s rRNA nucleotide sequences of Rickettsia-like endosymbiont (RLE), FLE, and viral protein sequences were retrieved from the GenBank database as indicated with the GenBank accession numbers in Figures 2 to 4. Sequences were aligned by using Muscle in MEGA-X software (54). Bayesian inference analysis was done using BEAST v1.10.4 software (55). Phylogenetic trees for the analysis of the 16s rRNA nucleotide sequences were constructed based on the Neighbor-Joining method with a pairwise deletion. The trees for the V3-V4 regions sequenced in this study were constructed with 500 bootstrap replicates (56–58) unless otherwise specified…”

Lines 157: The sentence was edited and completed, now can be found in Line 163: “… Neighbor-Joining methods were performed as an initial assessment for the identification and classification of the bacterial protein sequences and the…”

Line 159: “pulled out” was replaced by “retrieved” in the sentence and now can be found in line 168: “…protein sequences were retrieved from the GenBank database as indicated…”

Line 163 (After English editing this sentence can be found in Line 170): Yes, the four types of phylogenetic models were compared, and in the end, it was decided to accept the neighbor-joining method because it allowed the use of pair-wise deletion which allows sites containing missing data or alignment gaps to be removed from the analysis as the need arises.

Point 4: # Results

Line 208: If this Rickettsia is a true endosymbiont of D. nitens, wouldn't you expect it to be highly abundant in *all* samples, or at least all pools from the one region? In addition, it does not appear that your Rickettsial endosymbiont groups closely with other Dermacentor Rickettsial endosymbionts (Figure 2B). Is it possible that another tick species harbouring this rickettsia was accidently included into this pool?

Figure 2A & B. Please specify how many bp of the 16S rRNA gene was used to construct this tree. Many additional Francisella and Rickettsia sequences occur in GenBank - why were more not added to give a better picture of where these novel spp. fit into the phylogeny. Can you justify why the NJ cladogram is shown rather than the Bayesian or ML phylogram? How did the tree phylogenetic analyses compare? PCR and Sanger sequencing could have been done to generate more informative Francisella and Rickettsia sequences for phylogenetic analysis. Can you justify why this was not done?

Figure 4. Again, why are cladograms presented rather than phylograms?

Response 4: We appreciate your editings, all the suggestions were made in the following order and showed with the updated line number:

Line 215 (After English edited the manuscript the information contained in line 208 can be found now in Lane 215): Rickettsia is known as a common endosymbiont in some Ixodidae species but the information available about the endosymbionts hosted by Dermacentor spp. is scarce. Regarding the possibility that a different tick species was sequenced in one of the pools, This event is unlikely in this case because the ticks were taxonomically identified and then this classification was confirmed in consultation with the help of the NAMRU-6 collaborators.

Figure 2A & B.: the number of bp was specified in lines 223 and 227: “Francisella tularensis strains representing the phylogenetic relationship of 16S rDNA sequences (465 bp) OTUs classified as Francisella spp. in D. nitens. The tree was built using the pairwise deletion method. The blue branches represent the FLE clade, the green branches represent opportunistic pathogenic Francisella species, and the red branches represent the pathogenic Francisella tularensis strains as an outgroup. (B) Neighbor-joining cladogram rooted to pathogenic Rickettsia strains to represent the phylogenetic relationship of rickettsial 16S rDNA sequences (465 bp) with the OTU184 classified as Rickettsia spp. in the D. nitens sample”.

Can you justify why the NJ cladogram is shown rather than the Bayesian or ML phylogram? How did the tree phylogenetic analyses compare?: We decided to use the NJ method considering that this method permits to generate cladograms based on a pairwise deletions in the sequence comparisons among the NCBI sequences retrieved from the BLAST search.

PCR and Sanger sequencing could have been done to generate more informative Francisella and Rickettsia sequences for phylogenetic analysis. Can you justify why this was not done?: We agree with reviewer’s opinion for an accurate analysis. However, our goal in this study was to analyze the entire bacterial and viral composition in the blood-fed D. nitens female ticks. Therefore, categorization of our sequence to preexisting database was made, but not expanded to additional seqiencing. Our simlified approach may involve a bias since we would only have information about the main endosymbiont groups reported in ticks, but we would be missing all the other groups that were found by NGS targeting the V3-V4 regions of the 16S gene.

Figure 4. Again, why are cladograms presented rather than phylograms?

As our goal was only to describe if they were differences in the microbial composition between the different regions, we decided to present our results using cladograms to provide a condensed and simplified representation of evolutionary relationships based on gene sequences obtained to represent the relative relationship between taxa without the need for quantitative information about evolutionary distances of time.

Point 5: # Discussion

Line 344: This statement is misleading. Francisella endosymbionts are clearly phylogenetically distinct from F. tularensis.

line 345: Clarify that the ticks, and not F. tularensis is common in the northern hemisphere.

Line 407: Is there evidence that Chuviridae infect vertebrate hosts? If not, this hypothesis may not be correct.

Response 5: We appreciate your editings, all the suggestions were made in the following order and showed with the updated line number:

Lines 344: We reviewed and edited the sentence which now can be found in lines 352-353: “We found that the most abundant bacterium was FLE (80% of classified reads), which is phylogenetically distantly related to the pathogenic bacteria F. tularensis, and causes…”

Lines 345: Sentenced was edited and the suggested clarification was added and can be found in Lines 354-356: “While Dermacentor variabilis and Dermacentor andersoni, are known to carry this pathogen but are distributed in the northern hemisphere where F. tularensis is not commonly found, the effect of FLE interactions…”

Line 407 (After English editing the sentence can be found in line (423): By this sentence, we simply mean that vertebrate hosts can serve as a medium for the dispersal of tick species and their bacteria and/or viruses, since while these animals are marketed, they may be infested with ticks that will migrate as well and surely colonize other hosts. But to date, there are no reports of Chuviridae viruses infecting vertebrate hosts.

Round 2

Reviewer 1 Report

I do not have more comments to provide. The authors considered all my previous comments.

Reviewer 2 Report

All corrects have been made to a satisfactory standard.